# A Non-Destructive Detection and Grading Method of the Internal Quality of Preserved Eggs Based on an Improved ConvNext

**DOI:** 10.3390/foods13060925

**Published:** 2024-03-19

**Authors:** Wenquan Tang, Hao Zhang, Haoran Chen, Wei Fan, Qiaohua Wang

**Affiliations:** 1College of Engineering, Huazhong Agricultural University, Wuhan 430070, China; twq@webmail.hzau.edu.cn (W.T.); zzzzzzh66@163.com (H.Z.); 13966623103@163.com (H.C.); fanwei@mail.hzau.edu.cn (W.F.); 2Ministry of Agriculture Key Laboratory of Agricultural Equipment in the Middle and Lower Reaches of the Yangtze River, Wuhan 430070, China; 3National Research and Development Center for Egg Processing, Huazhong Agricultural University, Wuhan 430070, China

**Keywords:** preserved eggs, quality grading, ConvNeXt, multi-scale feature fusion, global attention mechanism, focal loss

## Abstract

As a traditional delicacy in China, preserved eggs inevitably experience instances of substandard quality during the production process. Chinese preserved egg production facilities can only rely on experienced workers to select the preserved eggs. However, the manual selection of preserved eggs presents challenges such as a low efficiency, subjective judgments, high costs, and hindered industrial production processes. In response to these challenges, this study procured the transmitted imagery of preserved eggs and refined the ConvNeXt network across four pivotal dimensions: the dimensionality reduction of model feature maps, the integration of multi-scale feature fusion (MSFF), the incorporation of a global attention mechanism (GAM) module, and the amalgamation of the cross-entropy loss function with focal loss. The resultant refined model, ConvNeXt_PEgg, attained proficiency in classifying and grading preserved eggs. Notably, the improved model achieved a classification accuracy of 92.6% across the five categories of preserved eggs, with a grading accuracy of 95.9% spanning three levels. Moreover, in contrast to its predecessor, the refined model witnessed a 24.5% reduction in the parameter volume, alongside a 3.2 percentage point augmentation in the classification accuracy and a 2.8 percentage point boost in the grading accuracy. Through meticulous comparative analysis, each enhancement exhibited varying degrees of performance elevation. Evidently, the refined model outshone a plethora of classical models, underscoring its efficacy in discerning the internal quality of preserved eggs. With its potential for real-world implementation, this technology portends to heighten the economic viability of manufacturing facilities.

## 1. Introduction

Preserved eggs, a cherished traditional delicacy in China, are not only celebrated for their potential therapeutic benefits in alleviating inflammation [1] and combating tumors [2], but also esteemed for their unique flavor and texture that captivate the palates of consumers. Crafted from fresh duck eggs, the process of pickling introduces a variety of factors such as the temperature, duck egg size, and concentration of the pickling solution, which inevitably yield varying degrees of quality in the final product. The grading standards set forth by the egg enterprises in central China categorize preserved eggs into three tiers: qualified preserved (QP) eggs, substandard preserved (SP) eggs, and inferior preserved (IP) eggs. Within the spectrum of SP eggs, notable classifications include pale preserved (PP) eggs, broken yolk preserved (BYP) eggs, and yellow yolk preserved (YYP) eggs. To ensure the integrity of the product, the preserved eggs undergo meticulous inspection and grading prior to packaging and distribution. Presently, the preserved egg industry relies heavily on manual visual scrutiny to identify and remove SP eggs and IP eggs. This labor-intensive process mandates the expertise of seasoned artisans, with a prerequisite of three to five years of experience, who meticulously evaluate the light transmission characteristics of preserved eggs under intense illumination. Despite its meticulous nature, manual inspection proves to be financially burdensome, operationally inefficient, and poses risks to ocular health due to prolonged exposure to bright lights. Moreover, subjective judgment and fatigue can compromise the accuracy of the inspection outcomes. To enhance operational efficiency and to propel advancements within the preserved egg industry, there is an urgent imperative for the development of automated technologies for the assessment of the internal quality and the grading of preserved eggs.

The detection of the internal quality in fresh poultry eggs has garnered significant attention in current research [3]. The leading detection methodologies encompass spectroscopic techniques [4], machine vision technology [5], and gas-sensitive sensors [6]. Cruz-Tirado et al. employed portable visible and near-infrared sensors to categorize egg freshness [7], while Liu et al. and Dong et al., respectively, utilized Raman spectroscopy and visible and near-infrared spectroscopy to predict the Haugh unit indicator of egg freshness [8,9]. Despite the high accuracy offered by the spectroscopic technology in detecting the internal quality of eggs, its exorbitant equipment costs and relatively sluggish detection speed have confined its utility primarily to scientific inquiry. In addressing the practical processing challenges within factories, cost-efficient and effective machine vision technology has emerged as the preferred solution. Jiang et al. leveraged GoogLeNet to automatically extract the features from speckled eggs, achieving a remarkable detection accuracy of 98.19% [10]. Aniyan et al. proposed a model incorporating an Android application scanner, utilizing machine vision technology for the non-destructive detection and grading of poultry egg freshness and age [11]. Recognizing the potential of machine vision in grading eggs’ internal quality, Liang et al. further employed this technology to enable the online detection of egg freshness and cracks [12].

Compared to the task of assessing the internal quality, detecting the external quality of preserved eggs is relatively straightforward. Existing research has explored non-destructive detection methods for spots [13] and cracks [14] in preserved eggs. Tang et al. further advanced this research by employing deep learning models to achieve the online detection of cracks in preserved eggs [15]. However, due to the limited translucency of preserved eggs, the complexity of factors affecting their quality, and the diverse nature of defective preserved eggs, the task of the non-destructive detection of the internal quality is considerably more challenging than assessing the external quality. Consequently, the literature on this subject is relatively scant. Chen et al. attempted to assess the gelatinous state of preserved eggs by analyzing their vibration signals [16]. However, due to the complexity of defective preserved eggs, merely identifying those with poor gelatinous quality is insufficient for comprehensive internal quality grading. Moreover, the collection of vibration signals is challenging and prone to inaccuracies caused by equipment vibrations, leading to a reduced detection accuracy. Wang et al. applied machine vision technology in conjunction with near-infrared spectroscopy to grade the internal quality of preserved eggs [17], demonstrating the feasibility of machine vision technology for the non-destructive detection of the internal quality in preserved eggs.

In recent years, propelled by the swift advancement of deep learning technology, a myriad of sophisticated models has emerged, enabling the resolution of myriad intricate image processing tasks. Consequently, deep learning has garnered widespread adoption in the realm of egg quality assessment [18]. Nasiri et al., by refining the VGG16 architecture, achieved the classification of pristine, bloodstained, and compromised eggs, attaining an impressive accuracy of 94.84%, surpassing conventional machine vision methodologies [19]. Turkoglu harnessed pre-trained residual network models to extract profound features, subsequently feeding these features into a bidirectional long short-term memory (BiLSTM) network, yielding a classification accuracy of 99.17% for images depicting soiled, bloodstained, fractured, and robust eggs [20]. Botta et al. employed convolutional neural networks (CNNs) to autonomously detect fissures in eggshells, achieving an accuracy of 95.38%, while also showcasing the superiority of the proposed CNN model in egg image classification over SVM models [21]. The aforementioned studies underscore the supremacy of deep learning models over traditional machine vision methodologies in egg image classification endeavors, promising to address the challenges that the conventional machine vision struggles to surmount. To this end, this study embraces ConvNext as the fundamental framework, enhancing the model’s detection efficacy through structural refinement, the incorporation of attention mechanism modules, and the optimization of loss functions, thereby tackling the arduous task of the non-destructive detection and grading of the internal quality of preserved eggs using machine vision technology.

## 2. Materials and Methods

### 2.1. Experimental Materials

Each category of the experimental samples was meticulously selected by experienced workers. Manual sorting involves illuminating the interior of preserved eggs under intense light to inspect their internal condition. The specific details of each type of preserved egg are illustrated in Figure 1:(1)IP eggs result from the pre-pickling eggs being spoiled or having cracks on the surface, leading to varying degrees of liquefaction inside, which causes the release of black water upon peeling and the emission of a noticeable foul odor. During visual inspection, IP eggs typically appear opaque, with no clear distinction between the yolk and albumen, while QP eggs exhibit translucency in the albumen and air chamber areas, with the yolk remaining opaque.(2)PP eggs arise due to various uncontrollable factors during the pickling process, resulting in preserved eggs not reaching the optimal pickling state, thus forming a type of SP egg. They mainly exhibit a poor gelatinous state and discoloration in the albumen. During manual inspection, PP eggs can be identified by their lighter translucency compared to QP eggs, often appearing yellow or pale yellow, whereas QP eggs generally exhibit a reddish or orangish-red translucency.(3)BYP eggs occur when the membrane of the preserved egg’s yolk ruptures, causing a partial leakage of yolk contents, resulting in an SP egg. During visual inspection, irregularities in the boundary between the yolk and albumen can be observed.(4)YYP eggs occur when fluctuations in the temperature during the pickling process prevent the yolk from completing the color transformation stage, resulting in a yellowish hue instead of the typical dark green. Through visual inspection, a yellowish tint at the edge of the yolk can be observed.

### 2.2. Data Acquisition

The image acquisition setup, as illustrated in Figure 2, comprised the following components: an AD-080GE dual-channel industrial camera from JAI, Copenhagen, Denmark (with a C-mount interface, a sensor size of 0.847 cm, a resolution of 1024 × 768 pixels, and a frame rate of 30 frames per second); an LM6NC lens from Kowa, Japan (compatible with the camera and featuring a C-mount interface); a Cob-brand lamp (with a power of 24 W, emitting warm white light); a rectangular dark box (with inner walls lined with black absorbent material); a baffle (covered with black absorbent material, with small holes in the middle); and preserved eggs placed on the holes.

This study amassed a total of 2127 images capturing various categories of preserved eggs. Among these, 692 images depicted QP eggs, 332 showcased PP eggs, 523 featured BYP eggs, and 257 displayed IP eggs. These images underwent partitioning into training, validation, and test sets, adhering to a 7:2:1 ratio, as detailed in Table 1.

Many advanced deep learning models rely on extensive datasets to mitigate the risk of overfitting. However, acquiring an adequate number of substandard and inferior preserved egg samples, as required for this study, posed a challenge due to their limited occurrence in practical production settings. To address this, data augmentation algorithms were employed to augment the dataset, thereby enhancing the model generalization and reducing the likelihood of overfitting, which often leads to improved accuracy [22]. Consequently, this study utilized a random assortment of augmentation techniques including horizontal flipping, vertical flipping, random rotation, random contrast adjustment, and random brightness adjustment. This augmentation process expanded the training dataset to ten times its original size, resulting in 14,870 augmented samples. The sample sizes of the validation and test sets remained unaltered. 

### 2.3. Model Improvement

#### 2.3.1. ConvNeXt

Liu et al., in their exploration of the architectural disparities between ConvNets and Transformers, discerned several pivotal components contributing to performance discrepancies. Building upon the foundation laid by ResNet and refining it through a series of enhancements, they introduced a pure convolutional model termed ConvNeXt. This model demonstrates competitiveness with state-of-the-art hierarchical Vision Transformers across various computer vision benchmarks, while still embodying the simplicity and efficiency characteristics of standard ConvNets [23], as depicted in Figure 3. ConvNeXt is classified into five versions—ConvNeXt_tiny, ConvNeXt_small, ConvNeXt_base, ConvNeXt_large, and ConvNeXt_xlarge—based on variations in the feature dimensions and network depth, with the corresponding model parameters delineated in Table 2. Here, d1 to d4 signify the reuse count of the ConvBext Block, corresponding to d1 to d4 in Figure 3, while dim1 to dim4 denote the dimensions of the output feature maps, corresponding to dim1 to dim4 in Figure 3.

Following preprocessing and feature extraction from hyperspectral images of pears, Liu et al. employed transfer learning to train the ConvNeXt model, successfully detecting early mechanical damage in pears [24]. Li et al. enhanced ConvNeXt and devised a classification model for ginseng grades, addressing the issue of low variability between features in ginseng grade classification [25]. Miao et al. amalgamated ConvNeXt with the ACMix network, augmenting ConvNeXt’s feature extraction performance and establishing a model for classifying traditional Chinese herbs [26]. These studies collectively substantiate the efficacy of ConvNeXt and its modified iterations in the domain of food detection. Therefore, this paper builds upon ConvNeXt, with the aim of refining its performance and tackling the classification of the internal quality of preserved eggs.

#### 2.3.2. Global Attention Mechanism (GAM)

Due to the predominance of opaque areas in preserved eggs, where the captured images predominantly feature black backgrounds with only a minor portion of translucent elements, this study’s algorithmic framework integrated the global attention mechanism (GAM) to enrich the representation of translucent regions within the preserved eggs. The GAM functions as an attention mechanism adept at diminishing information redundancy while magnifying global dimensional interaction features [27]. It embraces the Sequential Channel–Spatial Attention Mechanism extracted from the convolutional block attention module (CBAM) and refines its constituent sub-modules, as illustrated in Figure 4. Upon feeding a feature map *F*_1_ into two distinct attention sub-modules, the initial feature map underwent an initial correction via the Channel Attention Mechanism module, yielding an intermediate state denoted as *F*_2_. Subsequently, it underwent additional refinement through the Spatial Attention Mechanism module to yield the ultimate feature map denoted as *F*_3_. The Channel Attention sub-module, depicted in Figure 5, employs a three-dimensional configuration to preserve tri-dimensional information, subsequently employing a two-layer Multi-Layer Perceptron (MLP) to amplify cross-dimensional channel–spatial dependencies. The definition of the intermediate state is articulated in Equation (1), wherein *M_C_* symbolizes the channel attention map, and ⊗ signifies element-wise multiplication:(1)$$F_{2}=M_{c}(F_{1})⊗F_{1}$$

The spatial attention submodule, as illustrated in Figure 6, operates on the input feature map *F*_2_ with dimensions C × H × W. It employs two 7 × 7 convolutions to model the non-linear relationships among the pixels within the 7 × 7 blocks, thus enabling the parameters to capture more extensive spatial relationships among the pixels. The intermediate state is defined by Equation (2), where *M_S_* represents the spatial attention map:(2)$$F_{3}=M_{s}(F_{2})⊗F_{2}$$

#### 2.3.3. Loss Function

For the classification of the images, the prevailing loss function employed was the cross-entropy loss function, delineated by Equation (3):(3)$$L_{CE}=-\sum_{i=1}^{m}p(x)\log q(x)$$
wherein m denotes the batch size set for each iteration, i represents the i-th preserved egg image in the batch, *p*(*x*) denotes the true distribution probability of the five types of preserved egg images, and *q*(*x*) indicates the predicted distribution probability of the five types of preserved egg images. 

Due to the scarcity of SP eggs and IP eggs in the production process, sample collection presents challenges, compounded by the stochastic nature of occurrence probabilities across various types of SP eggs, rendering the attainment of a balanced sample collection arduous. Additionally, visually, the distinction between PP eggs and QP eggs is substantial, whereas the differentiation between BYP eggs and QP eggs is comparatively trivial, resulting in varying complexities in sorting different types of SP eggs. In response to this predicament, this present study adopts the focal loss function to mitigate the issue of imbalanced image data, as delineated in Equation (4):(4)$$FL\left(p_{t}\right)=-\alpha_{t}{\left(1-p_{t}\right)}^{\gamma }\log\left(p_{t}\right)$$
wherein *p_t_* represents the probability of the sample predicted as class *t*; *α_t_* and *γ* are hyperparameters; *α_t_* denotes the class weights, where adjusting its magnitude corresponds to the difficulty of class classification, balancing the uneven ratio of positive and negative samples; and *γ* is the adjustment coefficient in focal loss, which is aimed at reducing the weight of easily separable samples.

The focal loss mitigates the influence of class imbalance by diminishing the significance of the samples abundant in quantity while elevating the importance of those scarce in number. This strategic shift directs the model’s attention towards the more intricate specimens, thereby amplifying the precision and adaptability of the model. In this study, the conventional cross-entropy loss function was replaced with a hybrid of the focal loss function and the conventional cross-entropy loss function, depicted in Equation (5). This amalgamation empowers the model to concurrently prioritize both facile and arduous samples:(5)$$Loss=0.5\times L_{CE}+0.5\times FL$$

#### 2.3.4. ConvNeXt_PEgg

The previous investigation, building upon the ConvNeXt_tiny architecture, further diminished the dimensions of each module’s output feature maps from [dim1, dim2, dim3, dim4] to [64, 128, 256, 512] in order to reduce the model’s parameter count. Furthermore, enhancements were introduced across three key facets—the model structure, attention mechanism, and loss function—culminating in the proposal of a novel model tailored for classifying the internal quality of preserved eggs, denoted as ConvNeXt_PEgg, as depicted in Figure 7. 

Structurally, the adoption of multi-scale feature fusion (MSFF) brought about refinement. The output feature map from stage 4 underwent both upsampling and feature dimension reduction, transitioning the 7 × 7 × 512 feature map into 14 × 14 × 256. Subsequently, it merged with the 14 × 14 × 256 feature map output from stage 3, proceeding through fusion via three ConvNeXt blocks. Following the initial feature fusion, the resultant feature map underwent a subsequent round of upsampling and dimension reduction, transforming the 14 × 14 × 512 feature map into 28 × 28 × 256. This resultant map was then merged with the 56 × 56 × 256 feature map output from stage 2, undergoing further fusion via three ConvNeXt blocks. 

Deep networks inherently prioritize global features, whereas shallow networks emphasize local features. Through the amalgamation of feature maps derived from both shallow and deep networks, the model adeptly captured both local intricacies and overarching contextual information from the original images, thereby augmenting its recognition prowess and overall performance. 

Integration of the attention mechanism: A global attention mechanism module (GAM) was integrated after the outputs of stage 2 and stage 3 in the model, just before the fusion of the feature maps. Due to the relatively weak feature extraction capabilities of the shallow networks, the addition of the GAM module aimed to enhance the model’s focus on the translucent areas of the preserved eggs. 

Refinement in loss function: The fusion of the cross-entropy loss function with focal loss equipped the model with the capability to concurrently prioritize both straightforward and arduous samples, thereby enriching its discriminative capacity.

### 2.4. Model Training Environment

In this study, aimed at the non-destructive detection of the internal quality and grading of preserved eggs, a PyTorch framework was utilized to enhance ConvNeXt. To evaluate the detection efficacy of the different models and the effectiveness of the improvement algorithms, the training environment was standardized. The training platform comprised 128 GB of memory, equipped with an AMD Ryzen Threadripper 2920X CPU and an NVIDIA RTX2080ti GPU. The operating environment included a 64-bit Windows 10 operating system, Python 3.8 programming language version, PyTorch 1.9 deep learning framework, CUDA version 11.3, and PyCharm 2020.1 as the operating software.

### 2.5. Evaluation Metrics

Accuracy serves as a cornerstone metric of the classification model effectiveness, encapsulating the comprehensive performance of the classifier across all categories. A heightened accuracy signifies a greater proportion of accurately classified samples compared to the total sample size, indicative of the model’s superior holistic classification proficiency. The formal representation of accuracy is delineated by Equation (6):(6)$$Acc=\frac{TP+TN}{TP+TN+FP+FN}$$

The recall, or sensitivity, evaluates the classifier’s capability to accurately recognize positive instances. In multi-class scenarios, one class is designated as positive, while the rest are deemed negative. The formula for recall is represented by Equation (7):(7)$$Recall=\frac{TP}{TP+FN}$$

Precision is employed to assess how many of the predicted positive samples by the classifier are truly positive instances. The expression for precision is illustrated by Equation (8):(8)$$Precision=\frac{TP}{TP+FP}$$

True positive (*TP*) signifies instances correctly identified as positive among those that are truly positive; true negative (*TN*) indicates instances correctly identified as negative among those that are truly negative; false positive (*FP*) denotes instances incorrectly identified as positive among those that are truly negative; and false negative (*FN*) represents instances incorrectly identified as negative among those that are truly positive.

The F1-score amalgamates the precision and recall metrics, offering a holistic assessment of the model’s classification efficacy across each sample class, which is particularly adept in scenarios characterized by a class imbalance. The formulation for the F1-score is delineated by Equation (9):(9)$$F_{1}=\frac{2\times Precision\times Recall}{Precision+Recall}$$

Accuracy serves as a metric to assess the classification precision of all categories in binary or multi-class problems, being the most commonly employed evaluation criterion in classification models. However, it comes with limitations as it fails to evaluate the model’s classification performance for each sample class. To comprehensively assess the performance of the classification model, this study utilizes the accuracy to evaluate the overall performance, the F1-score to evaluate the classification performance for each sample class, and the parameter count to evaluate the model size.

## 3. Results

### 3.1. Training Results

This study compared the detection effectiveness of the improved model with different sizes of ConvNeXt models on the test dataset, presenting the classification results in Table 3. An analysis of Table 3 reveals that as the model size increases, the improvement in accuracy is minimal. In the context of the five-class classification task in this study, the largest model, ConvNeXt_xlarge, only achieved a modest 1.4 percentage point increase in accuracy compared to the smaller ConvNeXt_tiny model. In contrast, the refined ConvNeXt_Pegg model, developed using the proposed methodology, demonstrated a noteworthy 3.2 percentage point rise in accuracy compared to the original ConvNeXt_tiny model. Moreover, when compared to the largest ConvNeXt_xlarge model, it exhibited a 1.8 percentage point enhancement in accuracy. The data presented in the table underscore the significant challenge of distinguishing BYP eggs and YYP eggs among the five preserved egg categories, with all models yielding F1-scores below 90% for this classification. The enhanced model showed a considerable improvement in detection accuracy for the PP egg and IP egg categories, although the improvement was relatively minor for the BYP egg and QP egg categories. A slight decrease in detection performance was observed for the YYP egg category.

As the model used in practical applications is primarily aimed at classifying preserved eggs into three grades—PR egg, BYP egg, and YYP egg—that are all categorized as SP eggs, these three categories are consolidated into a single grade for the assessment of the accuracy and F1-score in the grading tasks. The outcomes are detailed in Table 4. An analysis of Table 4 reveals that the model’s size has minimal impact on its grading accuracy. The enhanced model developed in this study boasts the smallest parameter count among all models. In comparison to the smallest initial model ConvNeXt_tiny, the parameter count is reduced by 24.5%, accompanied by a 2.8 percentage point boost in the accuracy. Notably, its grading accuracy surpasses that of all other models, with a 2.4 percentage point increase over the largest model, ConvNeXt_xlarge. Furthermore, the F1-score demonstrates a significant enhancement in the detection efficacy across all three grades of the preserved eggs, underscoring the efficacy of the proposed improvement methodology outlined in this paper.

### 3.2. Validation Experiment

To evaluate the incremental impact of each enhancement on the model’s performance, this study systematically integrated each improvement method to observe its influence on the model’s classification and grading outcomes. These findings are presented in Table 5 and Table 6. Among these models, ConvNeXt_nano was derived from ConvNeXt_tiny by reducing the feature map dimensions, altering the original dimensions from [96, 192, 384, 768] to [64, 128, 256, 512]. ConvNeXt_MSFF was constructed by incorporating the MSFF structure into ConvNeXt_nano. ConvNeXt_GAM was formed by further integrating the GAM module into ConvNeXt_MSFF. Finally, ConvNeXt_PEgg emerged as the refined model, following additional adjustments to the loss function after ConvNeXt_GAM.

The analysis from Table 5 underscores that a significant reduction in the feature map dimensions has a marginal effect on the model’s classification accuracy, resulting in a mere 0.5 percentage point decrease. Among the array of enhancements, the integration of the MSFF structure manifests the most substantial impact on the model’s accuracy, elevating it from 88.9% for ConvNeXt_nano to 91.7%, translating to a notable gain of 2.8 percentage points. The incorporation of the GAM module and the refinement of the loss function yield relatively minor enhancements in the accuracy, with increases of 0.5 and 0.4 percentage points, respectively. The inclusion of the GAM module enhances the accuracy of the most challenging category, the BYP egg, while the adjustment of the loss function fortifies the accuracy of the small-sample categories like the PP egg and IP egg, particularly the IP egg, notwithstanding a slight decline in the detection efficacy for the YYP egg. This discrepancy might be ascribed to the pronounced visual contrast between the PP egg and IP egg, compounded by the limited sample size, which adversely impacts the detection performance of the model without the modified loss function. Subsequent to the refinement of the loss function, the model’s focus on these two categories intensifies, resulting in a significant enhancement in their detection efficacy. Conversely, the YYP egg exhibits minimal visual distinction, posing detection challenges and ultimately leading to no discernible improvement in the final detection performance.

From Table 6, it is evident that a substantial reduction in the feature map dimensions has only a minor impact on the model’s grading performance, resulting in a decrease of merely 0.9 percentage points. The MSFF structure emerges as the most significant enhancer of the model’s grading performance, showcasing an impressive increase of 2.7 percentage points. Both the incorporation of the GAM module and the refinement of the loss function contribute to a 0.5 percentage point improvement in the grading performance. Overall, all models exhibit a relatively robust grading performance for the QP egg with consistent results. The MSFF structure notably enhances the detection efficacy for the SP egg and IP egg, while the inclusion of the GAM module notably bolsters the detection efficacy for the QP egg, with a slight improvement also observed for the SP egg. Although there is a minor decrease in the accuracy for the IP egg following the addition of the GAM module, the refinement of the loss function results in a substantial improvement in the accuracy for the IP egg, with further enhancement observed for the SP egg. However, there is a certain degree of decline in the detection efficacy for the QP egg. Overall, the ConvNeXt_PEgg model maintains a high level of detection efficacy for all grades of preserved eggs, meeting the grading requirements in a factory setting.

### 3.3. Comparative Experiment

In order to thoroughly assess the superiorities of the refined model in the classification and grading tasks of the preserved eggs in this paper, the model presented herein will be compared with classic classification models such as DenseNet121, ResNet18, EfficientNet_b0, and MobileNetV3_large on this dataset. The classification and grading performances of each model are illustrated in Table 7 and Table 8.

From Table 7, it is evident that the most exemplary performance is observed with EfficientNet_b0, where its detection accuracy closely aligns with that of the unimproved ConvNeXt_tiny model. Conversely, MobileNetV3_large exhibits the poorest performance. Given that this model is designed for lightweight deployment on mobile devices, it experiences a slight decline in performance. Regarding specific categories, only ResNet18 and EfficientNet_b0 demonstrate slightly better detection efficacy for the QP egg and BYP egg, respectively, compared to ConvNeXt_Pegg. Overall, these classic models fall short in comparison to the improvements presented in this paper’s classification task. This could be attributed to the fact that these classic models were originally developed for public datasets without optimization for the characteristics of this paper’s dataset, thereby validating the success of the improvement methods proposed in this paper.

Based on the data presented in Table 8, it is apparent that these classical models possess notable advantages in terms of model parameters, particularly MobileNetV3_large, which boasts a compact size of only 3.0M. Nonetheless, their grading performance pales in comparison to ConvNeXt_Pegg. For applications requiring deployment on devices with restricted computational capabilities or mobile platforms, EfficientNet_b0 emerges as a viable option, with a modest model size of 4.0M, yet achieving a grading accuracy of 93.1%. Regarding the detection efficacy for each grade of preserved egg, only ResNet18 exhibits a comparable performance to ConvNeXt_Pegg in the case of the QP egg, while the other two grades significantly lag behind ConvNeXt_Pegg. This underscores that the model enhancements presented in this paper indeed furnish a superior solution to the challenges addressed herein.

## 4. Discussion

The proposed improved model ConvNeXt_Pegg demonstrates a significantly enhanced detection performance on the dataset compared to the previous ConvNeXt_tiny model, with notable increases in both the classification accuracy and grading precision. Specifically, the classification accuracy improved by 3.2 percentage points, while the grading accuracy rose by 2.8 percentage points. Each enhancement incorporated into the ConvNeXt_Pegg model contributed to varying degrees of improvement in the detection efficacy, with structural enhancements yielding the most prominent impact. This underscores the effectiveness of the enhancement methodology proposed in this paper. Furthermore, in comparison to other classical models, ConvNeXt_Pegg achieved the highest classification and grading accuracies. Compared to the top-performing classical model, EfficientNet_b0, ConvNeXt_Pegg boasted a 2.3 percentage point increase in the classification accuracy and a 2.8 percentage point increase in the grading accuracy. This notable improvement may be attributed to targeted enhancements tailored to the dataset, significantly enhancing its performance. Although ConvNeXt_Pegg exhibits a substantial reduction in the parameter count compared to ConvNeXt_tiny, it still lacks superiority in the parameter count compared to other classical models. Therefore, if stricter requirements on model parameters arise in the future, EfficientNet_b0 could serve as a foundational model, coupled with suitable improvement methods to achieve a detection performance close to ConvNeXt_Pegg while maintaining fewer parameters.

Preserved eggs undergo a pickling process where various factors, such as the quality of the raw eggs, the temperature, the concentration of the pickling solution, and the duration of pickling, exert their influence. These factors contribute to the distinctive characteristics observed in SP eggs during inspection, thereby significantly exacerbating the challenges of discerning the internal quality of preserved eggs. Given the unique nature of preserved eggs, the literature on the detection of the internal quality remains scarce. Wang and colleagues proposed a method that integrates machine vision technology with near-infrared spectroscopy to detect and grade the internal quality of preserved eggs [17]. In contrast to the singular utilization of machine vision technology for the internal quality detection and grading in this paper, Wang’s method entails a more complex detection process, necessitating multiple tests. Furthermore, it entails the drawback of the slow detection speeds and high equipment costs associated with near-infrared spectroscopy. Consequently, the approach presented in this paper emerges as more practical and offers greater utility in comparison.

## 5. Conclusions

This paper utilizes ConvNeXt_tiny as the foundational architecture and enriches it to yield the refined ConvNeXt_PEgg model tailored for the present dataset. This augmentation encompasses a reduction in the model parameters, the integration of the MSFF structure, the inclusion of the GAM module, and the amalgamation of the cross-entropy loss function with focal loss. These experimental findings unveil a classification accuracy of 92.6% and a grading accuracy of 95.9%. In contrast to the original model, there is a 24.5% reduction in the parameter count, accompanied by a 3.2 percentage point increase in the classification accuracy and a 2.8 percentage point increase in the grading accuracy.

The validation underscores the incremental contributions of each enhancement to performance enhancement. Particularly noteworthy is the MSFF structure, which yields the most substantial improvement, enhancing the classification and grading accuracy by 2.8 and 2.7 percentage points, respectively. Furthermore, the GAM module bolsters the model performance, while refining the loss function significantly enhances the detection efficacy for small samples. A comparative analysis demonstrates the superiority of the proposed enhancement methods over the classical classification models on this dataset, satisfying the grading accuracy requisites in practical production.

## Figures and Tables

**Figure 1 foods-13-00925-f001:**
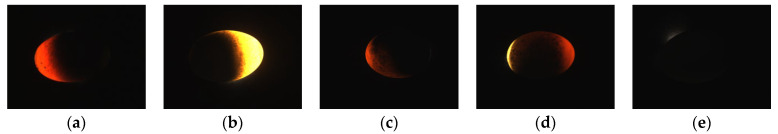
The preserved egg lighting inspection images: (**a**) QP egg; (**b**) PP egg; (**c**) BYP egg; (**d**) YYP egg; and (**e**) IP egg.

**Figure 2 foods-13-00925-f002:**
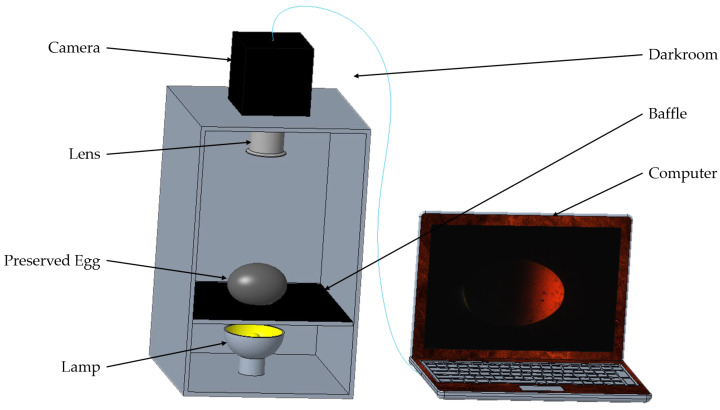
The preserved egg image acquisition device.

**Figure 3 foods-13-00925-f003:**
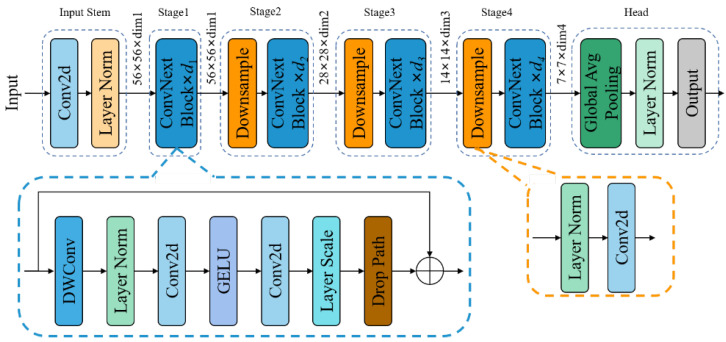
The network architecture of ConvNeXt.

**Figure 4 foods-13-00925-f004:**
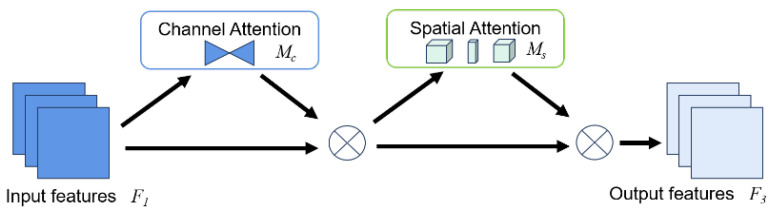
The overview of GAM.

**Figure 5 foods-13-00925-f005:**
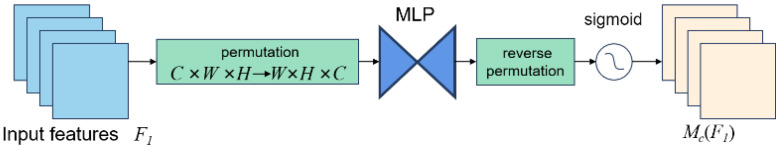
Channel attention submodule.

**Figure 6 foods-13-00925-f006:**
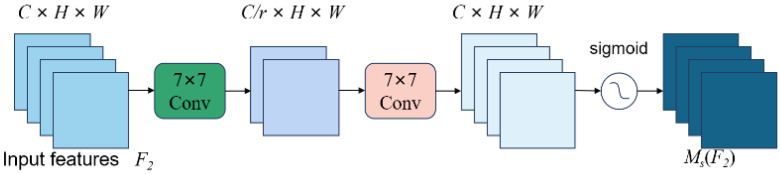
Spatial attention submodule.

**Figure 7 foods-13-00925-f007:**
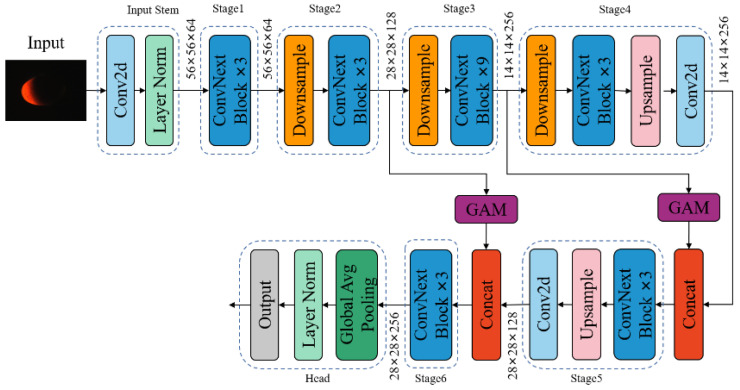
The network architecture of ConvNeXt_Pegg.

**Table 1 foods-13-00925-t001:** Distribution of the dataset.

Dataset	QP Egg	SP Egg			IP Egg	Total
		PP Egg	BYP Egg	YYP Egg		
Training set	484	232	366	226	179	1487
Validation set	138	66	104	64	51	423
Test set	70	34	53	33	27	217
Total	692	332	523	323	257	2127

**Table 2 foods-13-00925-t002:** Parameters of different versions of ConvNeXt.

Model	ConvNeXt_tiny	ConvNeXt_small	ConvNeXt_base	ConvNeXt_large	ConvNeXt_xlarge
Stage1	*d*_1_ = 3, dim1 = 96	*d*_1_ = 3, dim1 = 96	*d*_1_ = 3, dim1 = 128	*d*_1_ = 3, dim1 = 192	*d*_1_ = 3, dim1 = 256
Stage2	*d*_2_ = 3, dim2 = 192	*d*_2_ = 3, dim2 = 192	*d*_2_ = 3, dim2 = 256	*d*_2_ = 3, dim2 = 384	*d*_2_ = 3, dim2 = 512
Stage3	*d*_3_ = 9, dim3 = 384	*d*_3_ = 27, dim3 = 384	*d*_3_ = 27, dim3 = 512	*d*_3_ = 27, dim3 = 768	*d*_3_ = 27, dim3 = 1024
Stage4	*d*_4_ = 3, dim4 = 768	*d*_4_ = 3, dim4 = 768	*d*_4_ = 3, dim4 = 1024	*d*_4_ = 3, dim4 = 1536	*d*_4_ = 3, dim4 = 2048

**Table 3 foods-13-00925-t003:** Model classification results.

Model	F1-Score/%					Accuracy/%
	QP Egg	PP Egg	BYP Egg	YYP Egg	IP Egg	
ConvNeXt_tiny	93.8	94.3	82.4	87.9	86.3	89.4
ConvNeXt_small	92.4	90.1	84.9	88.9	89.8	89.4
ConvNeXt_base	93.6	94.4	85.4	89.2	86.8	90.3
ConvNeXt_large	93.2	90.4	87.8	86.2	92.3	90.3
ConvNeXt_xlarge	91.9	94.3	87.1	88.9	92.3	90.8
ConvNeXt_PEgg	94.4	95.8	88.2	87.9	98.1	92.6

**Table 4 foods-13-00925-t004:** Model parameter size and grading results.

Model	Parameter	F1-Score/%			Accuracy/%
		QP Egg	SP Egg	IP Egg	
ConvNeXt_tiny	27.8M	93.8	94.1	86.3	93.1
ConvNeXt_small	49.5M	92.4	94.2	89.8	93.1
ConvNeXt_base	87.6M	93.6	94.2	86.8	93.1
ConvNeXt_large	196.2M	93.2	94.1	92.3	93.5
ConvNeXt_xlarge	348.2M	91.9	94.9	92.3	93.5
ConvNeXt_PEgg	21.0M	94.4	96.2	98.1	95.9

**Table 5 foods-13-00925-t005:** Validating model classification results.

Model	F1-Score/%					Accuracy/%
	QP Egg	PP Egg	BYP Egg	YYP Egg	IP Egg	
ConvNeXt_tiny	93.8	94.3	82.4	87.9	86.3	89.4
ConvNeXt_nano	94.4	87.3	86.9	87.5	82.8	88.9
ConvNeXt_MSFF	94.4	93.0	86.5	90.6	94.3	91.7
ConvNeXt_GAM	95.8	93.2	88.2	89.2	92.3	92.2
ConvNeXt_PEgg	94.4	95.8	88.2	87.9	98.1	92.6

**Table 6 foods-13-00925-t006:** Validating model parameter size and grading results.

Model	Parameter	F1-Score/%			Accuracy/%
		QP Egg	SP Egg	IP Egg	
ConvNeXt_tiny	27.8M	93.8	94.1	86.3	93.1
ConvNeXt_nano	12.5M	94.4	93.3	82.8	92.2
ConvNeXt_MSFF	20.6M	94.4	95.4	94.3	94.9
ConvNeXt_GAM	21.0M	95.8	95.8	92.3	95.4
ConvNeXt_PEgg	21.0M	94.4	96.2	98.1	95.9

**Table 7 foods-13-00925-t007:** Comparative model classification results.

Model	F1-Score/%					Accuracy/%
	QP Egg	PP Egg	BYP Egg	YYP Egg	IP Egg	
DenseNet121	93.2	91.9	86.6	86.2	88.0	89.9
ResNet18	94.5	95.8	86.3	78.3	83.0	88.9
EfficientNet_b0	92.4	91.7	89.1	84.8	92.0	90.3
MobileNetV3_large	91.9	91.9	81.2	83.3	82.4	87.1
ConvNeXt_PEgg	94.4	95.8	88.2	87.9	98.1	92.6

**Table 8 foods-13-00925-t008:** Comparative model parameter size and grading results.

Model	Parameter	F1-Score/%			Accuracy/%
		QP Egg	SP Egg	IP Egg	
DenseNet121	7.0M	93.2	93.2	88.0	92.6
ResNet18	11.2M	94.5	92.7	83.0	92.2
EfficientNet_b0	4.0M	92.4	93.7	92.0	93.1
MobileNetV3_large	3.0M	91.9	91.1	82.4	90.3
ConvNeXt_Pegg	21.0M	94.4	96.2	98.1	95.9

## Data Availability

The original contributions presented in this study are included in the article; further inquiries can be directed to the corresponding author.

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
