# Peer review of "A Non-Destructive Detection and Grading Method of the Internal Quality of Preserved Eggs Based on an Improved ConvNext"

_foods, 2024, doi:10.3390/foods13060925_

Round 1
Reviewer 1 Report
Comments and Suggestions for Authors
Dear Authors,
The manuscript appears to be balanced, and written in readable English.
This is an interesting manuscript that deals with the current topic of vision systems combined withs machine learning methods for food quality assessment. The topic of the manuscript is original and of interest to the scientific community and the techniques used in the work are state of the art.
The introduction adequately covers the topic of the paper and the reason for the research carried out is clearly justified.
The methodology is clearly presented and the data, based on concrete tables, are suggestive. The data presented and discussed support the conclusions of the paper.
However, I have some suggestions which can help you to improve the work, please find them below:
1. In subsection 3.2. the authors describe four evaluation metrics, but in the results tables only two of them are presented. Why have the authors omitted recall and precision? I think that all metrics should be presented, as each provides important information about the quality of the models.
2. The authors present results for RP (or PR) eggs - Table 3, line 340 etc. But this group of eggs is not described in the methodology.
3. The structure of the manuscript is confusing. I think subsections 3.1 and 3.2 should be moved to section 2. Subsections 4.1 and next 4.1, which should be 4.2, are actually more comparative results of the ConvNeXt_PEgg model with other models and should be moved to section 3.
4. The manuscript lacks a discussion of the results obtained with the results of other researchers, which is an obligatory element of a good scientific paper.
Author Response
Thank you for taking the time to review our manuscript titled "Non-destructive Detection and Grading Method for Preserved Egg Internal Quality Based on Improved ConvNext," which was submitted for consideration for publication in Foods. We sincerely appreciate your valuable feedback and constructive comments.
We have carefully addressed each of the points raised in your review, and we provide our responses below:
Comment 1: In subsection 3.2. the authors describe four evaluation metrics, but in the results tables only two of them are presented. Why have the authors omitted recall and precision? I think that all metrics should be presented, as each provides important information about the quality of the models.
Response 1: Thank you for your valuable input. You are correct in noting that both recall and precision are indicative of the model's quality. However, recall and precision reflect the detection performance of the model for each class of samples. From this perspective, the F1 score combines both recall and precision, offering a more comprehensive assessment. Integrating recall and precision into the analysis could potentially lead to overly complex table data, hindering the clarity of the model's performance. Therefore, this paper solely utilizes the F1 score to evaluate the detection performance of the model for each class of samples, while accuracy is used to assess the overall detection performance of the model.
Comment 2: The authors present results for RP (or PR) eggs - Table 3, line 340 etc. But this group of eggs is not described in the methodology.
Response 2: Thank you for your valuable input. RP egg, standing for "rotten preserved egg," is indeed a category of IP egg. However, IP egg only encompasses this single category, hence there was no need for separate introduction. I inadvertently overlooked updating this detail in the text, which caused confusion. I have now replaced all instances of RP egg with IP egg in the manuscript.
Comment 3: The structure of the manuscript is confusing. I think subsections 3.1 and 3.2 should be moved to section 2. Subsections 4.1 and next 4.1, which should be 4.2, are actually more comparative results of the ConvNeXt_PEgg model with other models and should be moved to section 3.
Response 3: Thank you for your valuable input. Your suggestion has been immensely helpful, and it is indeed a great one. I have already revised the article structure according to your advice.
Commen 4: The manuscript lacks a discussion of the results obtained with the results of other researchers, which is an obligatory element of a good scientific paper.
Response 4: Thank you for your valuable input. Due to the significant challenges associated with detecting the internal quality of preserved eggs, there is limited research in this area. Among the existing studies, the work of Wang et al. is the most relevant to ours. Therefore, we have only compared our work with theirs in this paper. In comparison, our proposed method offers advantages such as simplicity in the detection process, faster detection speed, and lower equipment costs, making it more practical and valuable.

Reviewer 2 Report
Comments and Suggestions for Authors
All the comments are noted in the manuscripts' pdf file

Author Response
Thank you for taking the time to review our manuscript titled "Non-destructive Detection and Grading Method for Preserved Egg Internal Quality Based on Improved ConvNext," which was submitted for consideration for publication in Foods. We sincerely appreciate your valuable feedback and constructive comments.
We have carefully addressed each of the points raised in your review, and we provide our responses below:
Commen 1: Title of the work is appropriate
Response 1: Thank you for your recognition of this portion of the content.
Commen 2: Introduction section adequately describes scientific base for the research performed in this manuscript
Response 2: Thank you for your recognition of this portion of the content.
Commen 3: Provide literature citation for these statements
Response 3: In fact, there is currently scarce research on non-destructive detection of preserved egg internal quality, and there are no existing literature on preserved egg grading standards. The information presented in the paper is based on my actual investigation in the factory, hence there are no references available.
Commen 4: Materials and methods section provide sufficient information regarding applied equipment and modeling for the performed investigation
Response 4: Thank you for your recognition of this portion of the content.
Commen 5: Results and discussion sections are easier to track and comprehend if combined into one, integral section. Suggestion to the authors is to merge these sections into one
Response 5: Thank you for your valuable input, your suggestion has been immensely helpful and indeed insightful. Following your advice, we have incorporated Sections 3.1 and 3.2 into Section 2, while Sections 4.1 and 4.2 have been moved to Section 3. Additionally, we have rewritten the discussion section in Section 4 accordingly.
Commen 6: Conclusion section derives adequate conclusions from the presented results. Provide some directions for the furter research
Response 6: Thank you for your recognition of this portion of the content.

Round 2
Reviewer 1 Report
Comments and Suggestions for Authors
The authors corrected the manuscript according to my suggestions